

# Global transcriptome analysis of different stages of preimplantation embryo development in river buffalo

Chun-Ying Pang[1,*], Ming-Zhou Bai[2,*], Chi Zhang[2], Junhui Chen[2], Xing-Rong Lu[1], Ting-Xian Deng[1], Xiao-Ya Ma[1], An-Qin Duan[1], Sha-sha Liang[1], Yun-Qi Huang[3], Zhihui Xiu[2] and Xian-Wei Liang[1]

[1] Key Laboratory of Buffalo Genetics, Breeding and Reproduction Technology, Ministry of Agriculture and Rural Affairs (Guangxi), Guangxi Buffalo Research Institute, Chinese Academy of Agricultural Sciences, Nanning, P. R. China
[2] BGI Genomics, BGI-Shenzhen, Shenzhen, Guangdong, PR China
[3] Shandong Agricultural University, Taian, PR China
[*] These authors contributed equally to this work.

## ABSTRACT

**Background**. Water buffalo (*Bubalus bubalis*) are divided into river buffalo and swamp buffalo subspecies and are essential livestock for agriculture and the local economy. Studies on buffalo reproduction have primarily focused on optimal fertility and embryonic mortality. There is currently limited knowledge on buffalo embryonic development, especially during the preimplantation period. Assembly of the river buffalo genome offers a reference for omics studies and facilitates transcriptomic analysis of preimplantation embryo development (PED).

**Methods**. We revealed transcriptomic profile of four stages (2-cell, 8-cell, Morula and Blastocyst) of PED via RNA-seq (Illumina HiSeq4000). Each stage comprised three biological replicates. The data were analyzed according to the basic RNA-seq analysis process. Ingenuity analysis of cell lineage control, especially transcription factor (TF) regulatory networks, was also performed.

**Results**. A total of 21,519 expressed genes and 67,298 transcripts were predicted from approximately 81.94 Gb of raw data. Analysis of transcriptome-wide expression, gene coexpression networks, and differentially expressed genes (DEGs) allowed for the characterization of gene-specific expression levels and relationships for each stage. The expression patterns of TFs, such as *POU5F1*, *TEAD4*, *CDX4* and *GATAs*, were elucidated across diverse time series; most TF expression levels were increased during the blastocyst stage, during which time cell differentiation is initiated. All of these TFs were involved in the composition of the regulatory networks that precisely specify cell fate. These findings offer a deeper understanding of PED at the transcriptional level in the river buffalo.

Corresponding authors
Zhihui Xiu, xiuzhihui@genomics.cn
Xian-Wei Liang, liangbri@126.com

## INTRODUCTION

Water buffalo (*Bubalus bubalis*) is a species of the *Bubalus* genus in the Bovidae family; these buffalo are primarily distributed in tropical and subtropical regions (*Nanda & Nakao, 2003*). According to morphological and behavioral traits, water buffalo are classified into river buffalo and swamp buffalo (*Michelizzi et al., 2010*). As an important livestock source, the river buffalo provides draft power in agriculture and is a producer of milk and meat in several Asian countries (*Wang et al., 2017*). Although the river buffalo has clear economic importance in agriculture, its reproduction is still limited by long calving intervals, late puberty, and seasonal anestrus (*Guelfi et al., 2017*).

Studies concerning the reproduction of river buffalo have primarily focused on selection of optimal fertility through methods such as good nutrition condition and short-day fertility (*Campanile, 2007*; *Campanile et al., 2010*), with a goal of avoiding embryonic mortality, which is highly correlated withmaternal progesterone concentrations (*Neglia et al., 2008*). The mechanisms underlying preimplantation embryo development (PED) are unknown in the river buffalo. However, gene expression and regulation of the preimplantation embryo are critical for early cell fate decisions and transitions from totipotent to lineage specific cells (*Saiz & Plusa, 2013*). Transcriptomic profiling, either globally level or at the single-cell level, has been applied to the study PED in model animals (mouse and bovine) and in humans. As a result, expression patterns, transcriptional regulators (including transcription factors, lncRNAs and circular RNAs), and epigenomic reprogramming mechanisms have been revealed at representative stages of PED in these species (*Xie et al., 2010*; *Biase, Cao & Zhong, 2014*; *Graf et al., 2014*; *Fan et al., 2015*; *Guo et al., 2017*), resulting in successive waves of signal regulators and gene regulatory networks being established in PED (*Xie et al., 2010*; *Vassena et al., 2011*; *Zuo et al., 2016*). However, research on PED, even with respect to expression patterns and gene regulatory networks, is limited in the river buffalo.

Here, we report the global transcriptomic profiling for four stages of the preimplantation embryo (2-Cell, 8-Cell, Morula and Blastocyst) in the river buffalo. RNA-seq analysis from these four stages revealed gene expression patterns of the early phase of embryo development, characterized transcription in PED, and provided information on the regulatory networks involved in cell fate decisions and genetic control of early cell lineages. These results offer new insight into PED of the river buffalo.

## MATERIALS AND METHODS

### Ethics statement

Protocols were based on the Principles of the Administration of Experimental Animals issued by the Ministry of Science and Technology (Beijing, China) in 1988 (most recent version in 2001). The project was approved by the Institutional Review Board of BGI (NO. FT19030). All experiments were approved and supervised by the Animal Care and Use Committee of Guangxi Buffalo Research Institute.

### Oocyte collection and maturation culture in vitro

Based on a previously reported manual (*Liang et al., 2008*), ovarian follicles were collected from three different living river buffalos (Mora958, Mora1088 and Mora1172) by the

Ovum Pick-Up (OPU) method. Collected ovarian follicles were stored in the Storage Buffer (Table S3). Oocytes with more than three layers of granulosa cells were selected out under a stereoscopic microscope. After washing with the Storage Buffer three times, oocytes were incubated in glass plates with the Culture Medium (Table S3) at 39 °C and 5% $CO_2$ saturated air humidity.

## Preparation of granulosa cells

Per the description by *Liang et al. (2008)*, cumulus cells were eliminated from maturated oocytes after culturing for 22–24 h. Maturated granulosa cells were washed by the Culture Medium two times and suspended at a concentration of $\sim 2 \times 10^6$ cell/ml. A 30 µl microdroplet (10–15 maternal eggs) was plated into the plastic plate (35 mm diameter), covered with paraffin oil, and cultured for 30 min at 39 °C and 5% $CO_2$ saturated air humidity.

## In vitro fertilization (IVF) and zygote culture

Sperms were selected for 30 min in the Swimming-up Buffer (Table S3). Selected sperm were washed with the Fertilization Buffer (Table S3), and cocultured with mature maternal egg microdroplet in the Fertilization Buffer. The final sperm concentration was $1\sim 1.5 \times 10^6$/ml. The mixture was incubated for approximately 18 h. After eliminating sperm on the surface, zygotes were washed in the Embryo Culture Solution (Table S3) two-three times and transferred to single layer culture plates for incubation at 39 °C and 5% $CO_2$ saturated air humidity (*Pang et al., 2010*).

## RNA extraction and RNA-seq library preparation

Total RNA was extracted from zygotes at each stage by using Single Cell RNA-Seq library construction protocol (Illumina, San Diego, CA, USA). RNA integrity was asssessed via an Agilent Technologies 2100 Bioanalyzer. RNA-seq library preparation was performed as described previously (*Williams et al., 2017*). Seq-ready libraries were sequenced on the HiSeq 4000 platform (Illumina, San Diego, CA, USA).

## Data filtering and genome mapping

Raw reads were filtered by using SOAPnuke (Options = −l 10 −q 0.3 −n 0.05 −i) (BGI, Shenzhen, China). The criteria were as follows: (1) reads contain adaptors; (2) reads with unknown nucleotides $\geq$5%; and (3) low quality reads (the rate of reads whose quality value $\leq$ 10 is more than 20%). The river buffalo genome was used as a reference genome (*Williams et al., 2017*). Genome mapping was performed by using the HISAT (hierarchical indexing for spliced alignment of transcripts) method (Options = –phred64 –sensitive –no-discordant –no-mixed -I 1 -X 1000). Transcripts were reconstructed via StringTie (Options = −f 0.3 −j 3 −c 5 −g 100 −s 10000 −p 8) and predicted using Cuffcompare tool in Cufflink (*Trapnell et al., 2010*; *Pertea et al., 2016*).

## Gene expression and transcriptome-wide time series analysis

Clean reads were mapped to the reference genome via the Bowtie2 method (Options = −q –phred64 –sensitive –dpad 0 –gbar 99999999 –mp 1,1 –np 1 –score-min L,0, −0.1 −I 1 -X 1000 –no-mixed –no-discordant −p 1 −k 200) (*Langmead et al., 2019*). Gene expression

**Table 1  Overview of RNA-seq reads and mapping of the reference genome ($PE = 100$ bp).**

| Sample | Total Raw Bases (Gb) | Total Clean Bases (Gb) | Clean Reads Ratio (%) | Total Mapping Ratio (%) | Uniquely Mapping Ratio (%) |
|---|---|---|---|---|---|
| M958_2C | 6.36 | 6.16 | 97.22 | 77.65 | 63.78 |
| M958_8C | 7.01 | 6.22 | 88.74 | 84.54 | 65.68 |
| M958_BL | 6.90 | 6.53 | 94.68 | 66.76 | 51.85 |
| M958_MS | 6.79 | 6.18 | 91.11 | 83.17 | 62.62 |
| M1088_2C | 6.94 | 6.48 | 93.47 | 88.61 | 73.02 |
| M1088_8C | 6.97 | 6.48 | 92.89 | 82.54 | 67.74 |
| M1088_BL | 7.01 | 6.28 | 89.58 | 78.06 | 57.03 |
| M1088_MS | 6.96 | 6.46 | 92.79 | 85.06 | 67.84 |
| M1172_2C | 6.56 | 6.2 | 94.46 | 82.41 | 69.36 |
| M1172_8C | 6.95 | 6.18 | 88.9 | 82.71 | 67.82 |
| M1172_BL | 6.56 | 6.2 | 94.55 | 82.53 | 64.38 |
| M1172_MS | 6.95 | 6.38 | 91.75 | 85.76 | 67.78 |

levels were calculated using the RSEM (v 1.2.12) method (Options = –forward-prob 0.5) (*Li & Dewey, 2011*). Genes with similar expression patterns were clustered according to Mfuzz (Options = −c 12 −m 1.25) (*Kumar & Futschik, 2007*).

## Gene coexpression and differentially expressed genes

A gene coexpression network was constructed through WGCNA (weighted correlation network analysis) (threshold = 0.8, minModuleSize = 20, deepSplit = 2, power = 22). The differentially expressed genes (DEGs) analysis was performed by using DEGseq (fold change $\geq$ 2, adjusted $P$ value $\leq$ 0.001) (*Wang et al., 2010*) and PossionDis (fold change $\geq$ 2.00, FDR $\leq$ 0.001) (*Soneson & Delorenzi, 2013*). GO (gene ontology) terms for both co-expressed genes and DEGs were assigned according to the best-hits BLASTx, which were derived from Blast2GO (v2.5.0) alignments against the GO database (release-20120801). DEGs were aligned against KEGG (Kyoto Encyclopedia of Genes and Genomes) by the BLASTx package with $E$-value $\leq 10^{-5}$ as the threshold.

## RESULTS

### RNA-seq and mapping to the reference genome

To obtain transcriptomic information from different stages of PED in the river buffalo, we used the Illumina HiSeq platform to sequence twelve samples consisting four stages (2-Cell, 8-Cell, Morula and Blastocyst) with three biological replicates each. Approximately 81.94 Gb of raw data (bases) were generated. After filtering low quality reads, unknown nucleotides, and contained adapters from the raw reads, we obtained approximately 75.72 Gb of clean data (bases) with an average 6.31 Gb for each sample. Using the river buffalo genome as a reference (*Williams et al., 2017*), the average mapping ratio of each sample was ∼81.65%. Details were shown in Table 1. The quality of these RNA-seq data was satisfactory to perform further analysis, including gene expression statistics, transcriptome-wide time series expression profiling, coexpression and DEG detection and TF analysis.

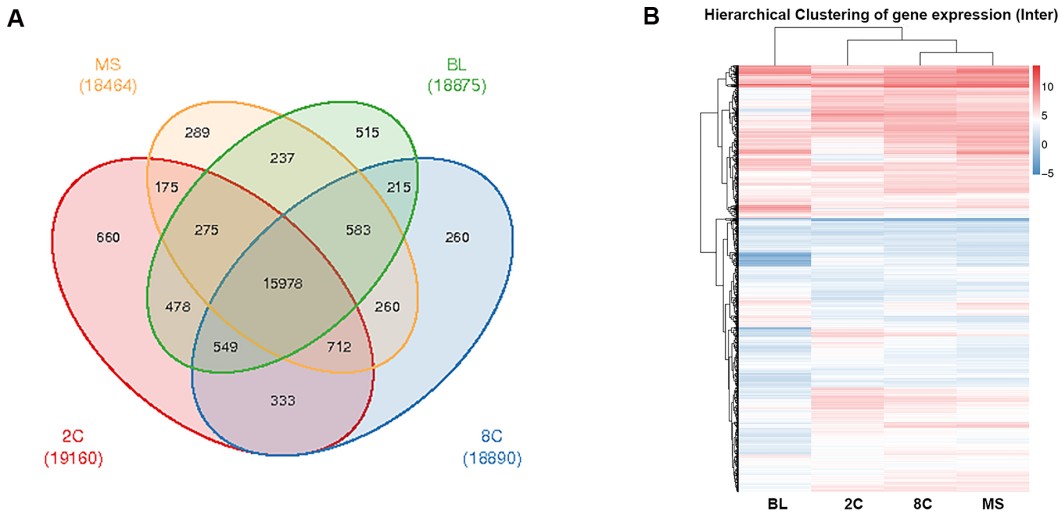

**Figure 1** **Gene expression statistics of river buffalo PED.** Gene expression statistics of four stages of river buffalo preimplantation embryo development. (A) Venn analysis of distinct and common genes expressed in four stages. (B) Cluster analysis of expressed genes via FPKM values for three replicates.

## Gene expression statistics

To elucidate gene expression profiles of the four stages of PED in the river buffalo, we annotated aligned reads based on the reference genome. The results revealed that 21,519 genes were expressed, which including 19,031 known genes and 2,488 novel genes. Among the 67,298 annotated transcripts, 28,136 transcripts with novel alternative splicing subtypes encoded known proteins, 2,512 transcripts were defined as novel protein coding genes, and 36,650 transcripts were classified into long non-coding RNAs. Details of each sample are shown in Table S1 and Fig. S1.

Venn analysis elucidated distinct and common genes expressed among the four stages of PED (Fig. 1A). This analysis revealed 660, 260, 289, and 515 specifically expressed genes in 2C (2-cell), 8C (8-Cell), MS (Morula), and BL (Blastocyst) stages, respectively. Furthermore, 15,978 genes were commonly expressed among these four stages. After filtering genes that had inconsistent FPKM values in the three biological replicates, we used mean FPKM values of the three replicates for cluster analysis (Fig. 1B). Cluster results indicated that functionally relevant genes showed similar expression patterns. In summary, combining transcript statistics of twelve samples with Venn and cluster analyses revealed gene expression profiles of the four stages of PED in the river buffalo.

## Transcriptome-wide time series expression profiling

Transcriptome-wide time series expression profiling is an efficient way to cluster genes that show similar expression patterns across different stages. To detect gene time series expression in four stages of PED, we employed Mfuzz to perform this cluster analysis (*Kumar & Futschik, 2007*) (Fig. 2). These clusters exhibited genes with specific expression patterns. In detail, clusters 1, 2, 3 and 4 comprised genes that were specifically expressed in 2C, 8C, MS and BL, respectively (Figs. 2A–2D). Genes in cluster 5 showed decreasing

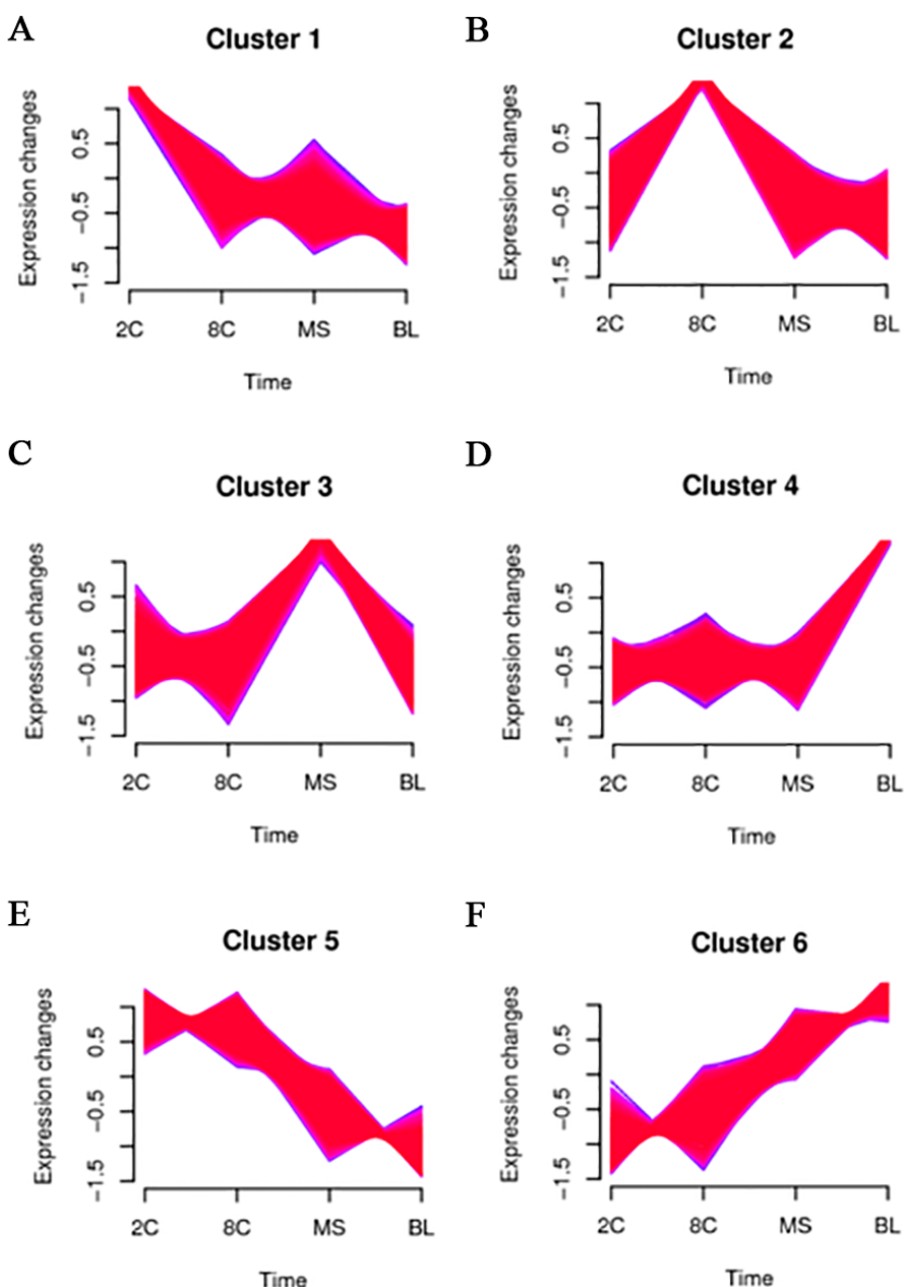

**Figure 2 Transcriptome-wide time series cluster of expressions profiling.** Transcriptome-wide time series cluster of expressions profiling through the different stages. (A–D) Genes that were specifically expressed in 2C, 8C, MS and BL, respectively. (E) Genes decreasing expression with PED progression. (F) Genes were increasingly expressed with PED progression.

expression with PED progression (Fig. 2E). In contrast, genes in cluster 6 were increasingly expressed with PED progression (Fig. 2F).

During PED, the expression levels of genes involved in the control of cell lineage are pivotal for embryonic development (*Saiz & Plusa, 2013*). In the time series expression

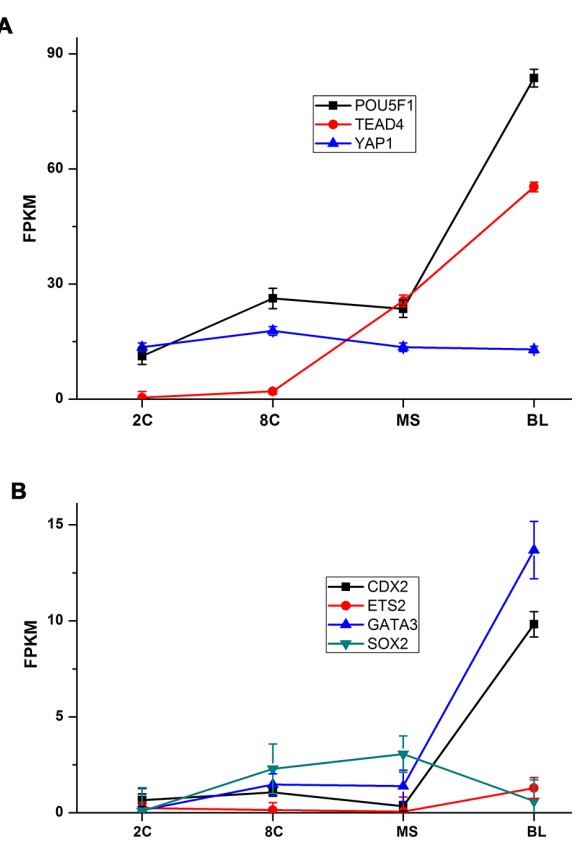

**Figure 3** **Ingenuity expression pattern analysis of specific genes involved in cell fate decisions.** Expression pattern of genes involved in cell fate decisions in preimplantation embryo development. (A) Genes with high expression levels (FPKM: 15–90) in all four stages. (B) Genes with relatively low expression levels (FPKM < 15). SD indicates three biological replicates.

profiling, we isolated genes reportedly involved in early cell fate decisions during embryonic development to perform ingenuity analysis (*Guo et al., 2010*; *Rossant, 2018*). Among them, *POU5F1* and *YAP1* showed high expression levels at the beginning of PED, and *POU5F1* showed a higher expression level during the BL stage (Fig. 3A). *TEAD4* began to show increased expression in the MS stage (Fig. 3A), while most of the other genes (except *SOX2*) showed an increased expression level in the BL stage (Fig. 3B). *SOX2* expression was upregulated in the 8C stage and peaked in the MS stage (Fig. 3B). These findings indicate that genetic control of cell lineage is activated during PED, and variant expression patterns might be due to the different roles of these genes in the regulatory network of cell fate decisions.

## Gene coexpression network analysis

We used WGCNA (weighted correlation network analysis) to detect gene coexpression networks at different stages of PED. Five modules were clustered by gene expression similarity and gene frac threshold (0.5) (Fig. S2). Nodes in the figure represent coexpressed genes, and line width indicates the coexpression relevance between genes. The larger the

node is, the higher the connectivity of the gene. Genes with high connectivity are located in the center of the network and play a key role in the whole module. Some genes with low connectivity but similar expression patterns are clustered as submodules (Fig. S2A). Such analysis provides information about key genes and their coexpression networks in PED (Table S2). We also performed correlation analysis between different modules (Fig. S2B). The relationship among different coexpression modules provides insight into horizontal levels of gene expression regulation (Table S2).

To gain information about the biological systems in which coexpressed genes participate, we performed gene ontology (GO) analysis of these coexpression modules (Fig. S3). In brief, cellular processes and biological regulation in the category Biological Processes, cell and cell parts in the category Cellular Components, and binding and catalytic activity in the category Molecular Function were significantly enriched in all coexpression modules. Such enrichments were highly correlated with active status during PED.

### Differentially expressed gene (DEG) detection

To distinguish DEGs at each stage, we used DEGseq (*Wang et al., 2010*) and PossionDis (*Soneson & Delorenzi, 2013*) to test for difference between each stage. The distribution of DEGs for each comparison is shown via MA plot (*Robinson, McCarthy & Smyth, 2009*) (Fig. 4A). Significantly up- and down- regulated genes are shown in Fig. 4B. All comparisons between the BL stage and any other stage (2C, 8C and MS) identified over 5000 up- and down- regulated DEGs, suggesting that gene expression during the BL stage is the most highly variable (Fig. 4B). 2C-VS-8C and 8C-VS-MS, two adjacent stage comparisons, exhibited fewer DEGs and fewer differences in expression levels (Figs. 4B, 4C). These findings indicate that gene expression profiling is more variable during the BL stage and less different between adjacent stages.

To illuminate the biological processes that participate in the four stages of the PED, we annotated the DEGs of each comparison according to both GO terms and KEGG pathway analysis. The top significantly enriched GO terms of DEGs in all comparisons were related to developmental processes, such as cellular and metabolic processes in Biological Process, cell/cell part in Cellular Component, and binding and catalytic activity in Molecular Function (Fig. S4). KEGG pathway annotation revealed the most dramatic variation occurred in metabolic pathway, which should be activated during cell division and differentiation (Fig. S5). Interestingly, pathways in cancer, which might result in cell compaction, were enriched in the 8C-VS-MS, at which time the embryo begins polarization during compaction (*Jedrusik, 2015*; *Niu, Mercado-Uribe & Liu, 2017*). These annotations suggest that processes and pathways related to development are activated in PED, reflecting pathways enriched in cell status transition between 8C and MS stages.

### Transcription factor (TF) analysis

Regulation of gene expression in PED is critical for cell division and differentiation (*Ma et al., 2001*). As transcriptional regulators, TFs play essential roles in the genetic control of early cell lineages (*Guo et al., 2010*) and suppose are thought to condition the distinct expression profiling during the four stages of PED. Therefore, we annotated TFs encoding

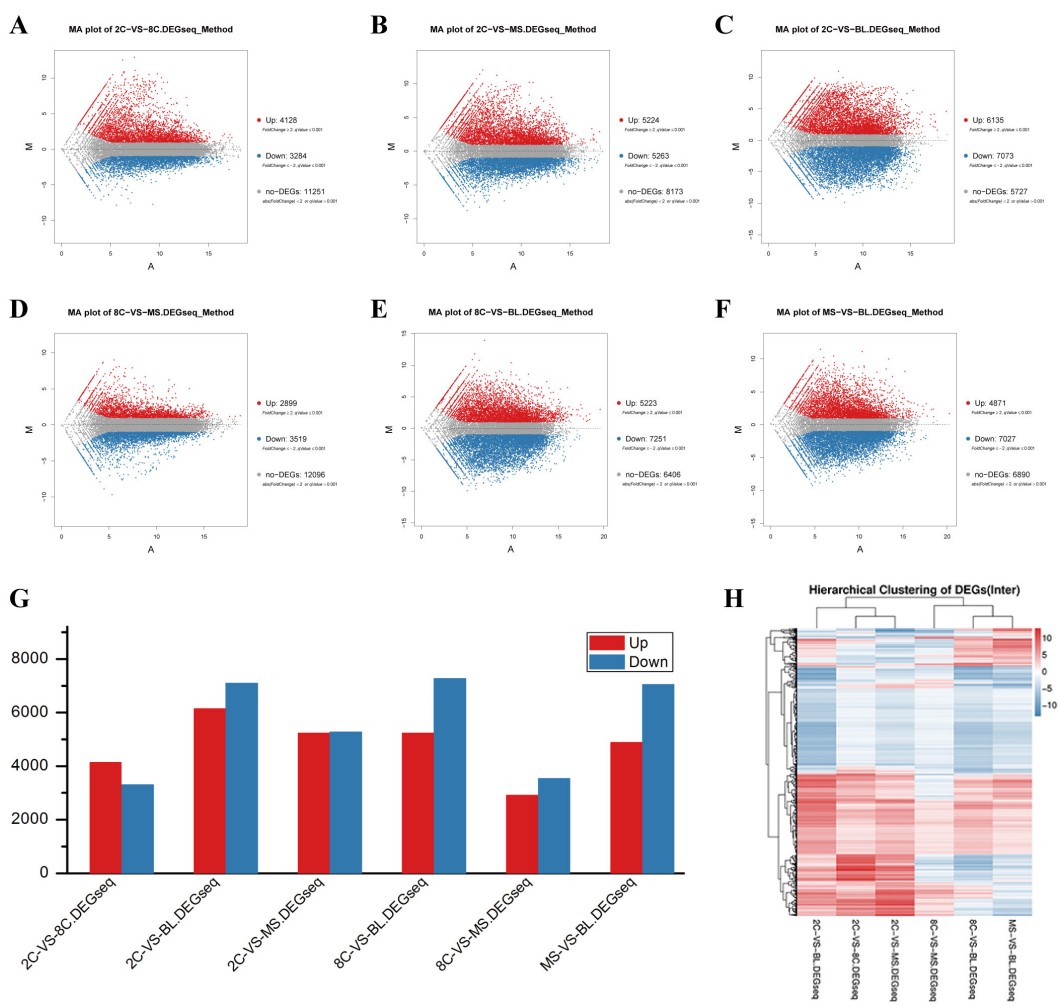

**Figure 4** **DEG analysis among four stages of PED.** Differentially expressed gene (DEG) analysis. (A–F) MA plot of DEGs in each two-stages comparison. (G) Up- and down- regulated DEGs in each comparison. (H) Cluster analysis of DEGs.

genes identified as DEGs and classified families of the TFs (Fig. 5A). Many families, such as the zinc finger and bHLH families, function in PED (*Jones, 2004*; *Guo et al., 2018*). Among these TFs, we isolated several predicted to be essential for cell fate decision for ingenuity analysis, unraveling their expression patterns according to the transcriptome-wide time series expression profiling (Fig. 5B). *GATA6*, *TCF7L1* and *JUNB* exhibited low expression levels in the 2C and 8C stages, increasing expression levels in the MS stage, and high expression levels in the BL stage (Fig. 5B). The expression levels of *NANOG* and *GRB7* were higher in 8C and MS stages than in the BL stage (Fig. 5B). *ATF7IP2* was highly expressed at the beginning of PED and maintained similar expression levels in the 2C, 8C and MS stages; however, *ATF7IP2* was downregulated in the BL stage (Fig. 5B). The expression patterns of these TFs were distinct at different stages of PED, suggesting that their differential expression is required during the control of cell lineage.

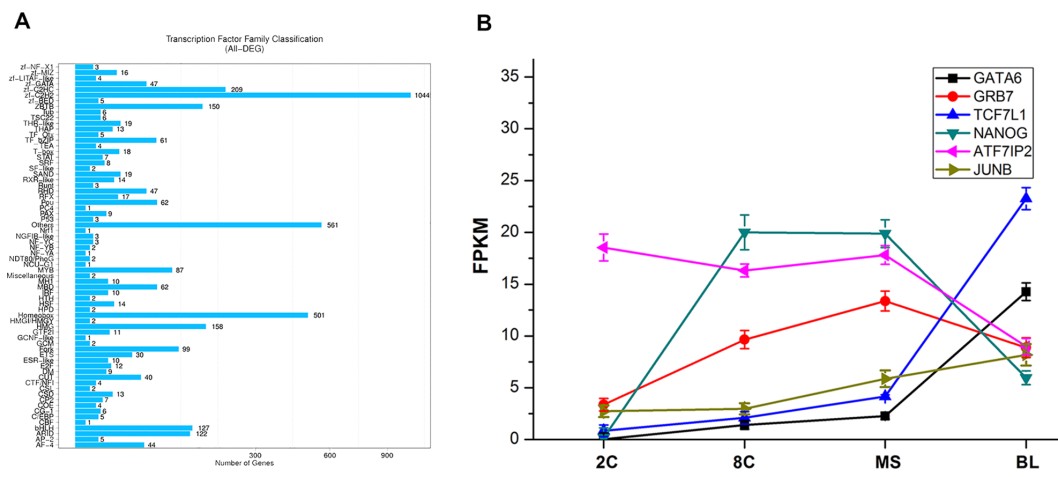

**Figure 5** **Differentially expressed transcription factor (TF) analysis.** DEG transcription factor (TF) analysis . (A) Statistics of TF numbers and families. (B) Time series expression of TFs involved in cell linage control. SD indicates three biological replicates.

## DISCUSSION

Preimplantation is the stage that initiates from zygote to blastocyst in mammalian (*Rossant, 2018*). The study of preimplantation embryos provides pivotal clues for genetic diagnosis (human), embryonic programming, and cell fate decisions (*Duranthon, Watson & Lonergan, 2008*; *Biase, Cao & Zhong, 2014*; *Wang et al., 2019*). Transcriptomic profiling of the preimplantation of humans and model animals has been performed both at the global transcriptome and single-cell transcriptome levels (*Huang & Khatib, 2010*; *Yan et al., 2013*; *Biase, Cao & Zhong, 2014*; *Jiang et al., 2014*). However, few studies are reported in buffalo. Our study unraveled the global transcriptomic profiles of PED in the river buffalo. Basic analysis of the RNA-seq data, which characterized the gene expression patterns, gene coexpression modules, and DEGs, not only creates a data foundation for further study of PED but also provides clues for the ingenuity analysis of specific pathways. For example, KEGG pathway analysis of the DEGs between different stages revealed that pathways in cancer were enriched, indicating that a few regulatory pathways in cancer are similar to those of PED. Such similarity, including ontogenesis and its crucial factors, was also reported in the comparison between the cancer stem cells and para-embryonic stem cells (*Manzo, 2019*).

Cell fate decisions in PED compose a complicated process that requires many regulators, such as microRNA (miRNA) and TFs (*Guo et al., 2010*; *Gurtan & Sharp, 2013*). Studies in mice have revealed lineage tracing and TFs involved in cell fate decisions (*Guo et al., 2010*; *Saiz & Plusa, 2013*). Among the numerous TFs, we chose key TFs for ingenuity analysis, revealing their expression patterns and analyzing possible genetic interaction hierarchy in the river buffalo (Fig. 6). In brief, *POU5F1(OCT4)/SOX2* (negative), *CDX2,* and *GATA3* are three parallel downstream effectors of *TEAD4/YAP1* (*Loh et al., 2006*; *Home et al., 2009*; *Ralston et al., 2010*). Negative feedback between *CDX2* and *OCT4* is essential for cell status

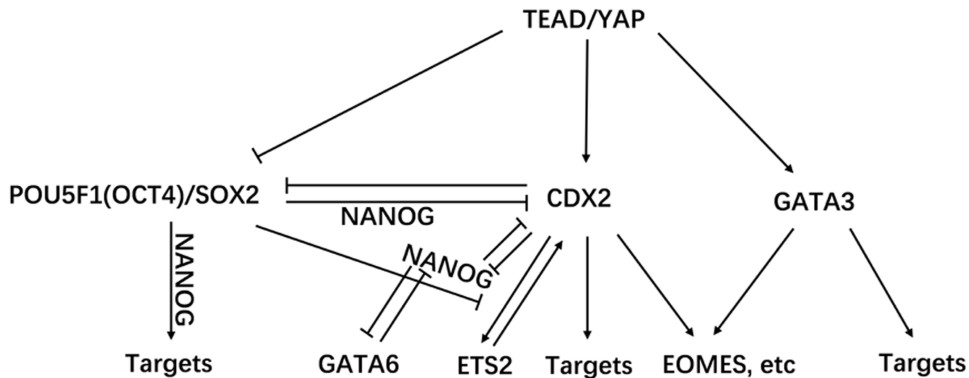

**Figure 6** **TFs regulatory network of cell lineage.** TF interaction hierarchy of cell lineage genetic control in river buffalo (*Rossant, 2018*).

stabilization (*Niwa et al., 2005*). *GATA3* shares downstream targets with *CDX2* (*Home et al., 2009*). The regulation between *ETS2* and *CDX2* is inhibited by *OCT4/SOX2* (*Rossant, 2018*). *NANOG* and *CDX2* repress each other and *NANOG* plays a subservient role to *OCT4* (*Chen et al., 2009*). *GATA6* and *NANOG* might repress each other's transcription (*Wamaitha et al., 2015*). Such regulation is reflected by their expression patterns in the river buffalo (Figs. 3 and 5). Along with the increasing expression level of *TEAD4*, expression levels of *CDX2* and *GATA3* were upregulated, and the *SOX2* expression level was downregulated. Subsequently, decreased inhibition of *SOX2* release the regulation between *ETS2* and *CDX2*, resulting in upregulation of *ETS2* expression. The expression patterns of *NANOG* and *GATA6* showed an opposite pattern. Interestingly, as the subservient role of *OCT4*, *NANOG* was expressed in a manner similar to the *OCT4* expression patterns during the 8C and MS stages, but did not increase in the BL stage. This difference may be due to the increasing expression level of *CDX2*. The expression level of *OCT4* was extremely high, supporting that *OCT4* is required for blastocyst formation (*Daigneault et al., 2018*).

Furthermore, expression patterns of other related TFs in PED indicated that they play roles in cell fate decisions in the river buffalo. In detail, *TCF7L1*, which mediate the Wnt signaling pathway, is necessary in pluripotent cells to prepare them for lineage specification (*Hoffman, Wu & Merrill, 2013*). *JUNB*, which belongs to AP-1 family and is involved in JAK-STAT signaling pathway, plays function in regulating gene activity (*Yamazaki et al., 2017*; *Yu et al., 2018*). The reason of their increased expression might be that cell division and differentiation are activated during the BL stage. *GRB7* encodes a growth receptor-binding protein that participates in various cellular signaling and functions (*Tai et al., 2016*). Its expression pattern showed high level at the 8C, MS and BL stages, which was consistent with previous report in mice (*Tanaka & Ko, 2004*).. *ATF7IP2* is a member of MCAF/AM proteins, which are able to interact with a variety of molecules to function as transcription modulators (*Fujita et al., 2003*; *Cai et al., 2006*). The high expression level of *ATF7IP2* might be due to the participation in the transcription regulation in PED. All of these TFs and other components compose a precisely regulatory network that controls cell lineages in river buffalo preimplantation embryo development.

## CONCLUSION

With RNA-seq data (∼81.94 Gb) from four stages of the PED in river buffalo, we characterized gene expression profiling, coexpression networks and DEGs. TF detection and transcriptome-wide time series expression analysis revealed TF expression patterns in the genetic control of cell linage. As regulators of cell fate decisions, TFs are involved in the construction of regulatory networks of cell lineage control.

### Funding

The study was funded by the National Natural Science Foundation of China (No. 31660649), Guangxi Science and Technology Major Project (GKAA16450002), and Guangxi Natural Fund program (2017GXNSFAA198108). The funders had no role in study design, data collection and analysis, decision to publish, or preparation of the manuscript.

### Grant Disclosures

The following grant information was disclosed by the authors:
National Natural Science Foundation of China: 31660649.
Guangxi Science and Technology Major Project: GKAA16450002.
Guangxi Natural Fund program: 2017GXNSFAA198108.

### Competing Interests

Ming-Zhou Bai, Chi Zhang, Junhui Chen and Zhihui Xiu are employed by BGI Genomics.

### Author Contributions

- Chun-Ying Pang performed the experiments, authored or reviewed drafts of the paper, approved the final draft.
- Ming-Zhou Bai and Yun-Qi Huang performed the experiments, prepared figures and/or tables, approved the final draft.
- Chi Zhang and Ting-Xian Deng analyzed the data, authored or reviewed drafts of the paper, approved the final draft.
- Junhui Chen analyzed the data, prepared figures and/or tables, approved the final draft.
- Xing-Rong Lu, Xiao-Ya Ma and An-Qin Duan performed the experiments, contributed reagents/materials/analysis tools, prepared figures and/or tables, approved the final draft.
- Sha-sha Liang analyzed the data, contributed reagents/materials/analysis tools, prepared figures and/or tables, approved the final draft.
- Zhihui Xiu conceived and designed the experiments, analyzed the data, prepared figures and/or tables, authored or reviewed drafts of the paper, approved the final draft.
- Xian-Wei Liang conceived and designed the experiments, analyzed the data, authored or reviewed drafts of the paper, approved the final draft.

### Animal Ethics

The following information was supplied relating to ethical approvals (i.e., approving body and any reference numbers):

Protocols were based on the Regulation on the Administration of Experimental Animals issued by the Ministry of Science and Technology (Beijing, China) in 1988 (last version in 2001). All experiments were approved and supervised by the Animal Care and Use Committee of Guangxi Buffalo Research Institute (BGI-IRB NO. FT 19030).

### Data Availability

The data are available at the CNGB Nucleotide Sequence Archive: CNP0000305.

### Supplemental Information

Supplemental information for this article can be found online at http://dx.doi.org/10.7717/peerj.8185#supplemental-information.

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
