# Peer review of "Global transcriptome analysis of different stages of preimplantation embryo development in river buffalo"

_PeerJ, doi:10.7717/peerj.8185_

## Round 0.1 · original submission · Major Revisions

Dear Dr. Xiu,

Please follow the essential comments by both Reviewer #1 and Reviewer #2. Reviewer #1 cites that "Although the authors provide great amount of data; it should be more descriptive and discussed as mentioned in the “Basic reporting” and “Experimental design” topics." Both reviewers also found that the references used for introduction were not appropriate and should be updated. Also data analysis, method presentation and discussion should be largely improved. For instance, both reviewers addressed several methods which could not be replicated if enough details are not given e.g. Reviewer #2 says " please list the parameters of WGCNA" and Reviewer #1 made similar comments about lack of methodological comments. Thus, please provide a detailed file responding all the comments.

Best regards
Rodrigo

Reviewer 1 ·

Basic reporting

The “Global transcriptome analysis of different stages of preimplantation embryo development in river buffalo (#37890)” is an interesting manuscript with a great amount of data for the early embryonic stages of the river buffalo. The introduction is concise and clear which integrates the reader into the question of the work. However, there are some relevant subjects I would like to address:
-The English language should be improved to facilitate the comprehension of text by an international reader. There are some professionals or companies that offer the language revision. The language could be improved for example, in the text lines: 25-26, 31-32, 53-54, 58-59, 155-158, 170, 173-174, 201, 206-207. Line 123 and 143 “three biological repeats” where “replicates” is a more appropriate term.
- The references should be reviewed. Line 54-55, for example is wrong. The text mention that “Global transcriptome profiling had revealed that gene expression of preimplantation embryo is essential for the embryonic and foetal development (Almen et al., 2009).” Almen et al 2009 refers to human membrane proteome. It does not mention embryo development or any stage of early development. Moreover, at Line 212-214 the authors start the discussion with a wrong information: “Starting from fertilized egg, the embryo undergoes a series of cell divisions to generate blastocyst which includes three distinct cell lineages—trophectoderm, inner cell mass and epiblast cells (Rossant and Tam, 2009).” The epiblast cells are derived from the inner cell mass. The third cell lineage is the hypoblast or primitive endoderm. The original information from Rossant and Tam, 2009 is: “The mouse blastocyst, immediately before implantation, consists of three distinct cell groups: the trophectoderm (TE); the epiblast, which is derived from the earlier inner cell mass (ICM); and the primitive endoderm”. Another example is in the discussion, line 255-257: “Its expression (GRB7) was along with the preimplantation embryo development and the down-regulation at BL stage indicated the EGFR didn’t keep as high level as in 8C and MS stage (Herrick et al., 2017).” Herrick et al 2017 does not refer to embryonic gene expression. In this paper the authors tested the influence of external growth factors in murine embryos cultured individually; “in a sequential medium system containing a defined protein source, blastocyst formation, hatching, and blastocyst cell number and allocation were not affected.” The authors could be more careful with their assumptions and the related literature. In this example they could focus more on GRB7 role itself, and not make indirect statements of its function.
- Regarding the references, along the manuscript the authors should use more recent papers from the literature. One example is in the discussion section, line 235, the description of POU5F1 function has only one reference from 1998. POU5F1 is an important transcription factor and has many papers related to its function. The authors could use more recent work as Daigneault et al 2018 (Scientific Reportsvolume 8, Article number: 7753) where they demonstrate that the “Disruption of the bovine POU5F1 locus prevented blastocyst formation and was associated with embryonic arrest at the morula stage”. This paper corroborates the data found in the present work and could enrich the discussion.
- In line 34, 124-125, 219, and 270, the authors say that 81.94Gb raw reads were generated and they show table1 to support this evidence. Table 1 does not have a complete subtitle and it makes difficult to understand this data, because if the total raw reads are given as megabases (M), the number of total raw reads should be 0.81Gb and not 81,94Gb. The same was told for the clean data that correspond to 75.72Gb and it can be 0.7572Gb.
-The authors could organize the figure2 in a more didactic manner. For example, all clusters with genes up-regulated would be together (e.g 1, 2 and 3), followed by the clusters with genes down regulated (4, 5, 6…), and finally add the clusters with variations of gene expression along development. In line 154 the authors say, “cluster 1 and 4 were genes which were up-regulated”. It is not easy to follow in the figure. If the clusters would be together, like clusters 1 and 2, it would be easier to read the text and search in the figure. Another example is “while cluster 6 and 8 showed down-regulated genes in 2C and BL”. These are two different clusters that show different profiles and could be separated in the explanation to give more details and explore other clusters. If they are talking about down-regulated genes at stage 2C, for example, they could have mentioned clusters 3, 7 and 11 as well.
- The Figure 3 should be described in more details in the text. The authors say that “All of them (genes) conditioned high expression in the preimplantation period and increased the expression (except SOX2) at the BL stage (Fig. 3a, b)”. Figure3 has two graphics, but they chose to talk only about the blastocyst stage. Moreover, the authors did not describe YAP1 expression, but talk briefly about it in the discussion (line245). A similar situation happened in the text describing figure 5b. The authors skipped the explanation for ATF7IP2 in the main text and in the discussion, although it has shown an interesting expression profile. I suggest the authors to comment more about their results in the main text related to Figures 3 and 5b.
- The “Gene co-expression network analysis” (line 166-177) findings is not discussed in the manuscript and the figures from this topic should be labelled, as Figure S2a and b. Table S2 refers to “5 Co-expression modules and genes in each modules”, but Figure S2a has 6modules containing two green modules. Labelling figures S2a and b might help the understanding of the results.
- The “Differentially expressed genes (DEGs) detection” topic is not discussed in the manuscript. It is only mentioned in the Discussion. This topic brings so much information and it deserves to be described in more details to be discussed later. Figures S3 and S4 have low resolution and it should be improved to facilitate the reading. The authors say that (line 193-195) “KEGG pathway annotation showed that the most dramatic variation in the six pairs of comparison was metabolic pathway that should be activated in cell division and differentiation (Fig. S4).” However, for 8C vs MS it shows pathways in cancer and transduction that could be described and discussed in the manuscript. This is quite interesting because the embryo starts its polarization during compaction and there are many papers in the literature, although in different mammals, that could help discussing the findings.
- The raw data was not supplied.
This work has generated an impressive amount of data, therefore they could describe and discuss more the information obtained from these results.

Experimental design

The manuscript is in the scope of PeerJ journal and intend to present new data regarding the transcriptome of buffalo preimplantation embryo stages. However the authors could improve the material and methods with details. Three topics (line 73-92): “oocyte collection and maturation culture in vitro”, “preparation of granulosa cells and in vitro fertilization (IVF)” and “zygote culture in vitro” do not mention “table S3 Formula of buffer in this study”. Moreover, the formula of the solutions in the table do not contain the exact concentration of the solutions, making difficult to replicate the experiment. Those topics (line 73-92) could be more detailed and should refer to any previous work if it has been done before.
The methodology could contain more details regarding data analyses, on topic “Data filtering and genome mapping (line 98-104)”.
On the topic “Transcriptome-wide time series expression profiling” (line 148-165), the authors “employed Mfuzz to perform this cluster analysis (line 151-152)” and found 12 clusters. They do not explain in the text all clusters present in Figure2. Clusters 3, 5, 6, 8, 11 and 12 are not included in the description of the results. If the authors do not consider them to be relevant, they could remove from the manuscript.

Validity of the findings

The manuscript present novelties in the gene expression profile of preimplantation embryonic stages of river buffalo with conclusion linked to original question of this work. Although the authors provide great amount of data; it should be more descriptive and discussed as mentioned in the “Basic reporting” and “Experimental design” topics of this review. Therefore, I recommend the article to be considered for publication after answering the revisions.

Reviewer 2 ·

Basic reporting

The preimplantation studies of embryos provide key insights into the question of when, where, and how cells take on fate separate. In this paper, the authors used RNA-seq to reveal the transcriptome profiles of four stages (2-cell, 8-cell, Morula and Blastocyst) of preimplantation embryo development. And gave an analysis of the gene co-expression network and differentially expressed genes (DEGs) between different stages characterized genes specific expression. Several TFs, POU5F1, TEAD4, CDX4 and GATAs, were also discussed. I believe it is an important work in deep understanding at transcriptional level about the preimplantation development. However, a careful revision is absolutely needed

Experimental design

Some parameters of WGCNA were not given

Validity of the findings

Discussion section is not like discussion, but like conclusion. So, the authors should rewrite this section.

Additional comments

The preimplantation studies of embryos provide key insights into the question of when, where, and how cells take on fate separate. In this paper, the authors used RNA-seq to reveal the transcriptome profiles of four stages (2-cell, 8-cell, Morula and Blastocyst) of preimplantation embryo development. And gave an analysis of the gene co-expression network and differentially expressed genes (DEGs) between different stages characterized genes specific expression. Several TFs, POU5F1, TEAD4, CDX4 and GATAs, were also discussed. I believe it is an important work in deep understanding at transcriptional level about the preimplantation development. However, a careful revision is absolutely needed as detailed below.

1. There are extensive reviews regarding transcriptome analysis in detection of embryo development signature, especially the preimplantation period. However, I failed to find sufficient novel insights to this well-established field. Please give the new findings or conclusions in their paper.
2. As I know, a lot of studies at transcriptional level about the preimplantation development were not given in the introduction. Please add the progress of early embryo from the transcriptome analysis[Vassena R, et al., Development. 2011; 138:3699-3709; Jiang Z, et al., BMC genomics. 2014; 15:756; Zuo Y, et al.. BMC genomics. 2014; 15:1113; Xie D, C et al., Genome research. 2010; 20:804-815; Biase F et al., Genome Res. 2014 24(11):1787-96; Yan L, et al., Nature structural & molecular biology. 2013; 20:1131-1139; Zuo Y, et al., Oncotarget. 2016 7(45):74120-74131; Hu B, et al., Open Biol. 2019 9(6):190054;Graf A et al., Proc Natl Acad Sci U S A. 2014 111(11):4139-44; Long, C, et al., IEEE Access, 2019 7: 7794-7802; Zhong S, et al., Nature, 2018, 555: 524-528; Fan X, et al., Genome Biology 2015,16: 148; Guo F et al., Cell Research, 2017, 27: 967-988, and so on], I believe this will be helpful for better understanding the research status of this area.
3. “Gene co-expression and differentially expressed genes” section: Please list the parameters of WGCNA: power, minModuleSize, deepSplit, neworkType 5 etc.,
4. The “Differentially expressed genes (DEGs) detection” section has similar question. The author used DEGseq and PossionDis to test the difference between each two stages, respectively, however the fold change, P value, Q value were not given.Lack of novelty.

---

## Round 0.2 · accepted · Accept

Dear Dr. Xu,

Reviewers recommended the publication of your manuscript since the text and figures were improved and all questions were answered and suggestions performed.

Reviewer 1 ·

Basic reporting

no comment

Experimental design

no comment

Validity of the findings

no comment

Additional comments

I recommend the publication of the manuscript: "Global transcriptome analysis of different stages of preimplantation embryo development in river buffalo". The authors made significant changes in the text and answered all the suggestions from the previous revision. They improved the text and figures and added a scheme as a final figure that made the manuscript interesting and easy to follow.

Reviewer 2 ·

Basic reporting

The English writing of thismanuscript has been imporoved.

Experimental design

no comment

Validity of the findings

This study is fully organized and systematically done

Additional comments

most of the comments have been revised in the new MS, I have no additional comment for this study.